# Comparative Investigation of the Critical Gap at Priority Junctions: A Review Paper

Mohammad Ali Sahraei

Department of Civil Engineering, College of Engineering, University of Buraimi, Al Buraimi P.O. Box 890, Oman; mohammad.a@uob.edu.om

**Abstract:** The crucial gap is an important aspect of traffic characteristics that is used to assess the delay and capacity of individual car movements at priority junctions. Because traffic operations at priority junctions are complicated, many methods have been studied to find a more accurate critical gap. This research examines the standards established for these procedures throughout the previous seven decades, from Raff's method in 1950 to the current day. These methods can be used anywhere in the world to determine the value of the critical gap for a mix of traffic, such as cars, vans, trucks, and motorcycles. The accuracy of these methods is assessed using factual data at two priority intersections, namely three-legged junctions (two-way stop-controlled (TWSC) with multilanes on main and minor streets). A total of 120 h of video camera recording was completed over the course of 5 working days in a week. This research identified Troutbeck's, Wu's, and Raff's techniques as the most popular, offering consistent and robust critical gap values for both sides of the minor road at priority junctions.

**Keywords:** critical gap; traffic characteristic; priority junction; capacity; delay

## 1. Introduction

At two-way stop-controlled (TWSC) junctions with two traffic approaches, vehicles streaming on the main street are prioritized for passing the junction; vehicles streaming on the minor street must wait for a sufficient gap on the main road. Tapio [1] explains that critical gap is the least noticeable flow headway within which a typical minor flow vehicle may maneuver. It is essential to assess the street's capacity and delay. The critical gap is a time criterion that determines whether a minor road vehicle may access the main route. Consequently, if the vehicle can access the main road when the headway on the main road is more significant than the critical gap, the headway is known as the "accepted gap"; if the vehicle is not able to access the junction when the headway is less compared to the critical gap, it is known as the "rejected gap". It is evident that several or identical drivers have several critical gaps due to various driving operations at different times. This acceptance gap difference is regarded as inconsistent and heterogeneous.

The critical gap is actually not a constant value in reality. It is actually a stochastic parameter using various magnitudes regarding several motorists as well as every motorist in various situations. Depending on the gap acceptance method, the motorist choice is constant in cases in which the identical gap size is generally recognized. The motorist's choice is considered homogenous if they recognize the identical gap length in similar circumstances [2]. In this regard, the lack of homogeneity and consistency results in several complicated methods. Numerous investigations by, e.g., Abou-Henaidy et al. [3], Ashworth and Bottom [4], and Kyte et al. [5] demonstrated that methods presuming the homogeneity and consistency of the driver's gap choice on the minor road differ from complex methods. For simplicity, most of the current techniques (e.g., Raff [6], Troutbeck [7], and Guo and Lin [8]) assume that the motorist's gap choice is homogeneous and consistent. Another issue which the gap acceptance method addresses is actually the distribution of practical

gaps within the main road that occur on the minor road, allowing cars to pass the junction or join the main road. Current methods vary based on the distribution of these gaps. For instance, the Transportation Research Board (TRB) [9–11], the New German Manual for Capacity of Priority Junctions [12], and a new Swedish capacity guideline [13] utilize negative exponential distribution, though the technique established in Australia utilizes Cowan distribution [14].

Since the value of the critical gap is crucial in terms of calculating control delay, performance measures, and levels of service (LOS) for the priority junctions, there are few studies reported, including Campisi et al. [15] and Tollazzi et al. [16]; thus, a gap can be found to determine the more precise method. In particular, the following research questions were produced so that good results for this research could be achieved:

What methods can be found for calculating critical gaps at priority junctions?

What are the magnitudes of the critical gaps based on the existing methods?

Which of the existing methods is the most accurate when it comes to comparing them?

Thus, the goal of the current study is to compare all methods in a thorough way to find the one that works best at priority junctions. Furthermore, in the current research, the methods were limited to the (three-legged formed) non-staggered and crossroads junctions. Consequently, the precision of these types of techniques was examined using factual information through two three-legged junctions as a form of priority junctions. In the literature, all methods for calculating critical gaps with their limitations and advantages are described in detail.

This research is actually arranged as follows. Section 2 presents a theory of critical gap followed by Section 3, which reviews the existing methods for calculating critical gap at priority junctions. A brief discussion about existing methods is provided in Section 4. Section 5 considers critical gap assessment and comparative analysis followed by the statement of the conclusion of this study in Section 6.

## 2. Theory of Critical Gap

With priority junctions with only two traffic streams, the gaps recognized by motorists on minor roads are evident. As shown in Figure 1, the passing timings of the main road vehicles in the conflict zone may detect gap occurrences. The length between two vehicles determines the duration of a gap. Nevertheless, in most situations, there are generally different traffic streams with varying priorities. Drivers with a lower priority must discover openings in the traffic caused by those with a higher priority. If the major road has several lanes, motorists on the minor road will only encounter gaps between these traffic streams, including the possibility of physical conflict. For instance, a minor road left turn (left-hand traffic) may encounter only right-side traffic gaps, while a minor road right turn (left-hand traffic) needs to consider all gaps between vehicles on the right and left sides [17].

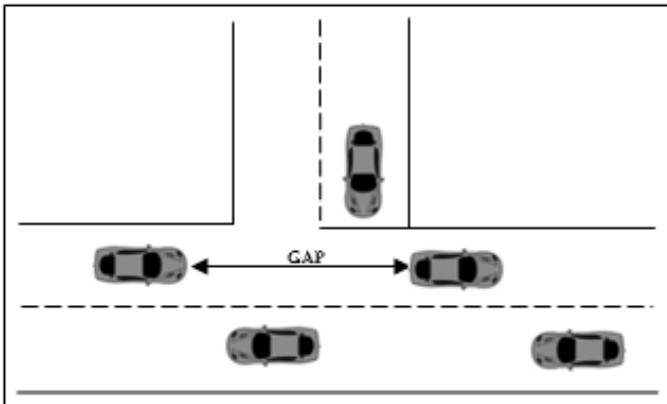

**Figure 1.** Visualization of the priority junction gap.

The critical gap is an essential parameter for delay evaluation at priority junctions. Since this parameter has a stochastic distribution, a site study is unable to directly obtain it. Evaluating critical gaps at priority junctions from traffic observation data is one of the most challenging tasks in traffic engineering. For calculating the critical gaps, statistical or empirical methods and procedures are needed. In reality, the gap distance differs from motorist to motorist as well as over time. It was discovered to differ depending on the subject vehicle type, approach gradient, junction geometry, weather conditions, and delay [18].

As one of the initial descriptions of the critical gap, Greenshields et al. [19] introduced the idea of the minimum average agreeable gap, a definition of the lag accepted by more than 50% of motorists. Later, Raff [6] introduced the idea of critical lag, which refers to the duration of the lag and has the feature that the number of accepted lags is less than it is equal to the number of refused lags larger than it. According to the TRB [9–11], the critical gap is defined as the minimum period, in seconds, between successive main road vehicles in which a minor road vehicle can generate a maneuver. Therefore, the value of the critical gap is the smallest allowed gap. A particular motorist might decline several gaps lower compared to the critical gap and might also approve gaps identical to or even higher than the critical gap value. As a result, critical gaps for a given junction may be computed depending on observations of the approved gaps as well as the largest declined gaps.

Regarding the subject mentioned above, different methods exist for calculating critical gaps at priority junctions, as described in Section 3.

## 3. Methods of Calculating Critical Gap at Priority Junction

### 3.1. The Raff's Method

This strategy was initially proposed by Raff [6], and due to its simplicity and effectiveness, it remains one of the most used strategies for unsaturated circumstances. The original Raff's technique approximated key lags based on lags accepted and rejected, which was deemed statistically inefficient since it excluded all the gap data [20]. As described by Brilon et al. [21], or by combining gaps and lags [22,23], Raff's approach was expanded to calculate important gaps either by taking into consideration just gaps or by combining gaps and lags.

Based on Raff's description, crucial gap is actually the length of the gap where the set of gaps shorter than it and the set of gaps greater than it are equal. This is shown by the intersection of the (Fa) and (1-Fr) curves, where (Fa) and (Fr) represent the cumulative probability of approved and refused gaps, respectively.

Miller [24] found that this method works well enough with a small amount of bias when there is not much traffic because it only uses lag data. However, it is statistically inefficient when there is a lot of traffic because it ignores many useful data points. Tupper et al. [25] found that when only lag data was used, this method produced results with a large error. Because all of the declining gaps are taken into account in this method, careful drivers are overrepresented. To eliminate such data bias, some specialists have examined only the largest gaps that have been rejected.

### 3.2. Solberg' Method

Solberg and Oppenlander [26] used a probit evaluation to establish a statistical procedure for estimating the critical gap. In this regard, the rejection or approval of a specific period gap is displayed by a binomial response, or all-or-nothing; the length of the gap influences this. This method for the predicted percentage of accepting a period of the gap is given in Equation (1):

$$Y = 5 + 1/\sigma(X - \mu) \tag{1}$$

where X = the logarithm of the time gap, $\sigma$ = the standard deviation of tolerance distribution, $\mu$ = the mean tolerance distribution, and Y = the probit of the percentage accepting time gap.

Solberg and Oppenlander [26] pointed out that there was no substantial distinction between the lag and gap acceptance times; therefore, lag and gap are able to be used together. Nevertheless, additional research by Wagner [27] discovered that critical gaps acquired using only gap acceptance periods were generally overestimated and that the utilization of lag information only led to data loss.

### 3.3. Ashworth's Method

Ashworth [28,29] developed a technique to include gaps and lag sizes in his research. For this investigation, gaps and lags size were identical and assumed to comply with an exponential distribution. In this regard, Ashworth extracted a formula for calculating the values of critical gaps as a function of the mean approved gap size and the variance of the approved gap size. The result in Ashworth's research is valid only if minor street vehicles are assumed to arrive at the "Give Way" line independent of the arrival of the following main street vehicle.

To test the validity of this technique under conditions where this independence of arrivals for main traffic flow was assumed, computer simulation methods were used with an infinitely long queue of vehicles on the minor street. A constant "move-up time" of 3 s was assumed to signify the minimum headway between two minor street vehicles accepting the same main street gap. In addition, Ashworth and Bottom [4,28] noted that motorists who had long waiting times accepted a shorter gap time than other people.

Based on Ashworth's method, Miller [24,29] presented another modified method for estimating the critical gap at the priority junction and assumed that the main street followed a gamma distribution. This is a very sensible description and can be utilized to give good results.

### 3.4. Adebisi's Method

Adebisi and Sama [30] considered the effect of the stopped delay experienced by minor street motorists on their gap acceptance behavior. The analysis concentrated on left-turning minor street traffic (right-hand driving system) and was restricted to conditions with small or no queue development. The sites studied were two TWSC junctions in Nigeria. Both areas are along the same main street, and the main street is a two-lane two-way with a median about 1.5 m wide on average. No big trucks or buses were observed during the analysis. Data were collected on a typical working day between 9:00 am and 12:00 pm. The data extracted from the field study were classified based on delays experienced by the left-turning minor street traffic, such that motorists who experienced delays of lower than 5.0 s were in the same class, 5.1–10.0 s were in the next class, and so on. Eight such categories were identified for motorist delays of up to 40.0 s, while those who experienced delays of 40.1–60.0 s and higher than 60.0 s were also categorized into two different groups.

Adebisi and Sama [30] computed the mean and variance of the investigated drivers' crucial gap using the computer program (CHOMP) [31–33]. All rejected and acceptable gaps were utilized in the computer software mentioned above. The analysis was first conducted on the aggregated data before moving on to the disaggregated data. The mean crucial gaps obtained for each type of halted delay were then statistically compared to those obtained from the data.

The average crucial gaps (i.e., 10.44 s obtained) from the aggregated data were more significant than the values indicated in past studies: 5.3 s [34–36], 6.0 s [33], and 5.2 s and 7.74 s [37] for average halted delays smaller than 25 s, according to Adebisi and Sama [30]. Even if minor street drivers experience delays that are more than 30 s on average, they tolerate smaller than usual gaps. Furthermore, Adebisi and Sama [30] discussed how the consequences of poor driver judgment are possibly more severe in small gaps. As a result, when minor street traffic average delays surpass 30 s, some kind of control, such as signalization, would be appropriate to improve safety. Adebisi and Sama [30] emphasized that while the investigation's evaluation was done using both aggregate and disaggregate data, only unsaturated circumstances on the minor roadway were looked into.

### 3.5. Troutbeck's Method

Research by Brilon et al. [21] described how the model of Troutbeck [7] provides the best outcomes. Therefore, this method is suggested for calculating the critical gaps in TRB [9] and TRB [10]. In order to apply the maximum probability approach to establish a driver's critical gap, the driver's critical gap must lie between the range of the maximum rejected gap and approved gap. It should be considered that the critical gaps have a probabilistic distribution. The Troutbeck model [7] is a microscopic approach, and the strategy is based on the notion of maximum likelihood. Only the maximum declined gap and the acceptable gap for a single vehicle are taken into consideration in this procedure. Additionally, two suppositions are made: (1) that the crucial gaps have a log-normal distribution and (2) that the behavior of the motorists is uniform and consistent. According to Brilon's research [38], the Troutbeck approach [7] is extremely difficult to apply and has weak results. For this procedure to produce consistent results, a sizable sample size is also required. Brilon [38] used a hyper-Erlang distribution in the aforementioned approach. It was said that both procedures yielded comparable outcomes.

The Implementation of the Maximum Likelihood Method by Tian

Using field data, Tian et al. [39,40] focused on applying the maximum likelihood approach to determine a motorist's critical gap (right-hand driving system) by using accepted gaps and the maximum rejected gaps. They specifically mentioned how all of the present techniques call for the same information, such as accepted gaps and denied gaps. The accepted gap and the maximum rejected gap for each driver are necessary for the maximum likelihood method. No study has been conducted on how to extract accepted and declined gap data from field data or how to detect gap events when several traffic streams are actually present. Without a detailed description of the gap occurrences, it is impossible to implement the different methods for computing the critical gap from field data. Using the computer technique created by Kyte et al. [5], the gap event data from the site study were recovered in this case. The program provides a data file that contains a listing of all the gaps that each minor road automobile has accepted or not accepted. In addition, the data file is analyzed by a computer that extracts all approved and refused gaps for a given movement. The data are then used as input for the program developed by Troutbeck for determining maximum likelihood [7]. The program reveals the standard deviation, the range of data, and the average critical gap.

The research looks at how the critical gap method is used to handle major road right-turn movements and multilane scenarios. In this method, when searching for gap occurrences at intersections with more than two lanes, vehicles in the lane without a conflict are not taken into account. This leads to a wider critical gap than at single-lane junctions. A larger critical gap would arise in the right-turn movement if the main road right-turn movements were not taken into account when determining gap events. On the other hand, if major road right-turning cars are taken into account while identifying gap events, the critical gap is reduced. In both circumstances, the gap acceptance procedure is misunderstood [39,40]. According to Tian et al. [39] and Xu and Tian [41], one solution to the aforementioned issue is to account for the influence of right-turn movements when computing conflicting flow rates, as the Highway Capacity Handbook (HCM) specifies. Remarkably, the right-turn effect is not taken into account when determining the critical gap. In specific circumstances, such as a two-stage gap acceptance procedure, pedestrian obstruction, driver lane preference, and downstream queue spillback, the suggested method cannot be applied directly.

### 3.6. Pant's Method

Research by Pant and Balakrishnan [42] explained the improvement of a binary-logit technique and an artificial neural network (ANN) technique for predicting approved and declined gap times at low-volume and rural TWSC junctions. The information was gathered at 16 priority junctions on rural Ohio highways. The speed limit on the main street was about 55 mph. Furthermore, only one (1) lane existed at each junction direction, and there was no individual lane for left-turn movements (right-hand driving system). In order to do data collection, a video camera recording technique (i.e., six video cameras) was used for 3–4 h throughout the night and day at each TWSC junction. Four cameras were installed next to the minor street and two alongside the main street to record all vehicle movements near or at the junction. The data were examined by reviewing the videotapes accordingly.

In the case of data analysis, the data were separated into two parts. One set of the data was used for modeling, and the other set was used to validate the models. The form of control, the existence of a vehicle in the opposite approach, a queue in the minor street, vehicle speed, service time, the length of the gap, and the turning movements on both the minor and main street were identified to impact the motorist's selection to approve or refuse a gap size. In this regard, the outcomes of the binary-logit method and ANN were investigated with the actual data sets in the site study. Accordingly, the outcomes showed that the ANN properly forecasted a more significant proportion of accepted or declined gaps than the binary-logit method [42].

Pant and Balakrishnan [42] described the significance of this research in that although the ANN created in the above research was dependent on data gathered in a low-volume rural area at a TWSC junction, the method could be utilized for establishing new models for TWSC junctions in urban regions with approximately high traffic volumes.

### 3.7. Tian's Method

Tian et al. [40] conducted research to identify elements that might influence the critical gap: for example, junction geometry, traffic movements, vehicle type, city size, speed limits, average delay, rural junction vs. urban junction, and approach grade. The analysis was carried out using the linear regression technique to indicate the importance of these different elements. Such an evaluation is a macroscopic assessment approach because the critical gap values were acquired by observing a range of motorists. In this case, a macroscopic database related to the objective was created. This included calculations of the critical gap at more than 40 priority junctions in 5 geographical regions all over the United States. In order to determine the essential parameters that can influence the critical gap, a regression analysis was carried out by excluding or including specific parameters. Then, a variance evaluation was performed for a comparison between regression equations for both conditions, i.e., excluded or included parameters, as well as to find whether or not the parameter had a substantial effect on the critical gap. In this regard, overall observations of the several elements can undoubtedly be described as follows:

- The main elements influencing the critical gap were identified as junction geometry (e.g., three-leg or four-leg, single-lane or multilane), movement type, type of vehicle, vehicle delay, and approach grade;
- By increasing the level of traffic on the main route or automobile delays on the minor street, motorists often find shorter gaps;
- By increasing the number of legs at junctions or the lane number on the main road, the critical gap value can increase;
- By increasing the road grade, the critical gap can grow as well;
- Critical gaps for heavy vehicles were found to be consistently more significant than those for other vehicles.

### 3.8. Pollatschek's Method

Pollatschek et al. [43] and Polus et al. [44] provided a microscopic decision model for motorist gap acceptance behavior (right-hand driving system) when they waited on a minor street at a priority junction. This research evaluated the values of the critical gap for all categories of motorists. In recording the characteristic behavior of motorists, he described how almost all motorists are generally assumed to be consistent and homogeneous in their behavior, and their values of gap acceptance performance cannot change with time. Nevertheless, the above presumption is incorrect because of variations among motorists, such as driving manners, age, and gender. Furthermore, distinct individual motorists have varying patience levels toward risk when they are waiting in a queue. Additionally, risk tolerance is based on the waiting period in the queue and behind the stop line. In particular, Pollatschek et al. [43], Polus et al. [44], and Polus and Pollatschek [45] asserted that the new model is actually relying on the evaluation of the risk connected with declining small gaps alongside the possible advantage from acceptance, that is the time period saved because of smaller waits behind the stop line.

Pollatschek et al. [43] classified two categories of motorists, careful and risk-loving, who had different behaviors. Consequently, if the number of risk-loving motorists increases at a priority junction, many vehicles can enter the junction per unit time compared to the population, which includes a majority of careful drivers. Although increasing the number of risk-loving drivers results in the acceptance of a smaller gap, it presents an additional risk to the general entry process. Pollatschek et al. [43] asserted that the design values derived for gap acceptance from some countries could not create representative values in other parts of the world.

### 3.9. Yan's Method

Researchers Yan et al. [46] and Yan and Radwan [47] examined how driver gender, driver age, and main traffic speed (right-hand driving system) affect gap acceptance behaviors. The study's findings showed connections between drivers' choices for the left-turn gap, their rates of acceleration and steering control, and how the length of gap acceptance affected other cars in the main traffic lane. The study looked at two main road traffic speed scenarios, two driver genders, and three age groups independently. The age ranges for the young, middle-aged, and elderly groups were 20–30, 31–55, and 56–83, respectively. The University of Central Florida (UCF) driving simulator was also used in the study. Both scenarios—40.2 km/h on the main roadway and 88.5 km/h on the main street—were tested on every participant.

The research outcomes specified that motorists are more likely to accept shorter gaps at higher main street traffic speeds than at lower main street traffic speeds. For the gender variable, the outcomes demonstrated that male motorists accepted shorter gaps than female motorists, representing an overall conservative driving attitude of female motorists. As a result of their diminishing driving skills, the elderly drivers displayed a cautious driving stance. Therefore, the consequences of perceptual and cognitive ageing may result in generally vulnerable people in relatively complicated driving maneuvers, particularly for older female motorists [46,47].

Although Yan et al. [46] and Yan and Radwan [47] demonstrated that a relationship among main street traffic speed, driver gender, and age with the gap acceptance maneuver could be demonstrated, the authors did not perform field validation for the simulator data. Therefore, caution must be observed in transferring these results directly to practice in absolute terms. In addition, Yan et al. [46] described how both significant highway design guides, including the American Association of State Highway and Transportation Officials [48] and the TRB [9], ignored the results of significant elements, including motorist age and gender and main street vehicle speed, on gap acceptance, which have usually been given attention by experts in traffic safety aspects.

### 3.10. Sangole's Method

According to different drivers and traffic characteristics in India, Sangole et al. [49] and Patil and Sangole [50,51] derived the right-turning maneuver behavior for two-wheelers at TWSC intersections using the minor road gap acceptance method. An adaptive neuro-fuzzy interface system (ANFIS), which provides an optimization structure to find the fuzzy system variables that best match the data, was created in this instance using the MATLAB software. During typical workdays, video camera recordings were used to collect data at three priority intersections throughout the morning (10 am–11 am) for roughly 60 min at each junction. The main street featured four lanes at all intersections (i.e., dual lanes in each direction). A video camera was placed on the terrace of a building with a great view of all three entrances to the traffic stream. To gather information on drivers, including estimated age and gender and vehicle occupancy, a video camera was positioned at road level.

Based on this study, Sangole et al. [49] described how India's stop or yield signs do not function. As a result, the control of cars at priority junctions is actually challenging as well as extremely interactive, and all motorists must make specific choices regarding where, when, and how to execute a necessary move. Such junctions are not able to be analyzed based on the results of priority junctions in other countries.

In this study, the various variables gathered consisted of vehicle arrival rate, gap approved/declined, type of conflicting vehicle, period to cross the junction, approximate driver age and gender, and occupancy of two-wheelers. Next, eight different models with various combinations of parameters were considered, and then the prediction capability of models was compared with that field data set. The parameter conflicting vehicle type could not be identified with much efficiency in the model prediction, while the motorist's age was found to be the significant variable in gap acceptance selection [49–51].

Sangole et al. [49] and Patil and Sangole [50,51] both agreed that no research has previously focused on how two-wheelers behave while accepting gaps at TWSC crossroads. Understanding two-wheeler behavior is crucial to the development delay approach for priority junctions. In addition, this study concentrated only on right-turning maneuvers from minor roads at TWSC junctions; therefore, more research should include other turning movements and four-legged junctions with more vehicle classes (passenger car, heavy vehicle, bus).

### 3.11. Guo's Method

Guo and Lin [8] and Guo et al. [52] proposed a method for determining critical gaps. This model was based on the probability density function of a declined and accepted gap on a minor road and a bunched exponential distribution of headway (M3) on a major road. This method was created based on the initial presumptions at a priority junction, in which the main and minor roads are both one-way traffic flows. For simplicity, Guo and Lin [8] assumed the following specific conditions: 1. independence between the arrival times of the minor road vehicles and the main road vehicles; 2. motorist behavior that is both homogeneous and consistent.

Guo and Lin [8] and Guo et al. [52] demonstrated some variations between the methods, such as the maximum likelihood technique, Hewitt's technique, and Raff's technique. These procedures calculate the realistic value of the critical gap based on field samples. In Guo's technique, a measurement of the critical gap is theoretically acquired and comprises the procedure of gap acceptance theory. Raff's technique is only a specific case of Guo's technique. In addition, they described how additional investigation is required to calibrate the values of coefficients for the improved gap acceptance method based on large sample sizes in different types of priority junctions.

Furthermore, depending on Raff's description of the critical gap, Guo et al. [52] and Guo [53] created a method referred to as the revised Raff's technique. The method suggested the presumption of independence between minor road vehicles' arrival times and those of main road vehicles. New methods were tested by simulating headway data and assessing different critical gap techniques, such as Ashworth's and Raff's technique.

The summary shows that the revised Raff's method obeys the description of Raff's method, and the application is actually additionally reliable compared to Raff's method. It is capable of producing a more exact result than Raff's approach. Furthermore, Ashworth's technique requires presenting a rigorous presumption that the approved gap as well as critical gap carry out the standard distribution, while it is challenging to provide the presumption regarding the site study or simulated information. Eventually, Guo et al. [52] and Guo [53] asserted that the revised Raff's method has a minor fluctuation, while Raff's technique and Ashworth's technique include more significant variation within various circumstances.

### 3.12. Wu's Method

A technique for assessing critical gaps at priority junctions was presented by Wu [54,55]. This novel methodology's theoretical underpinning is actually the probability equilibrium among the accepted as well as declined gaps. Using the cumulative distribution of the denied and accepted gaps, the equilibrium is macroscopically determined. Similar outcomes are obtained by the new approach and Troutbeck's [7] model.

Interestingly, the technique's feature that all rejected gaps, not only the largest declined gap, may be considered is the primary differentiation among the innovative approach as well as the most popular Troutbeck technique [7]. The novel technique produces results (deviations lower than 0.2 s) regarding the mean critical gaps similar to those through the Troutbeck technique [7] in cases where only the greatest declining gaps are used. The computed mean critical gaps ought to be smaller than Troutbeck's technique if all declined gaps are used. Wu [54,55] suggests a suitable computing approach for using the suggested macroscopic method. A spreadsheet may be used to complete this operation rapidly (for instance, EXCEL or QuatroPro).

According to Wu [54,55], the new method has the following advantageous characteristics: (a) a theoretically sound foundation; (b) robust results; (c) independence from any method assumptions; (d) ability to take into account all relevant gaps; (e) ability to directly obtain the empirical probability distribution function of the critical gaps; and (f) a straightforward computation with no iteration. Additionally, it does not need the consistency or homogeneity of drivers, nor does it require a fixed distribution function of the critical gaps.

### 3.13. Devarasetty's Method

By taking into account several possibly influencing factors at a priority junction, Devarasetty et al. [23] devised a binary logit approach to determine the likelihood of rejecting or approving a particular lag or gap regarding a left-turn maneuver on a major road (right-hand driving system). The approach's significant elements were found using a stepwise selection procedure. In this respect, a logistic regression approach was created for which the independent parameters consist of traffic characteristics and the geometry of a field, and the dependent parameters consist of the values of gap/lag acceptance of a left-turning maneuver from the main street. For a more accurate assessment of traffic operations, this approach may be used to determine the magnitudes of the critical lag or gap regarding a given combination of traffic flow and site characteristic factors. In addition, gap and lag acceptance data for left-turning movements from a main street were obtained from over 30 field trials in the US. Every field of study has a stop-controlled intersection at the minor street.

Two individual models, including lag and gap models, with substantially different variables were created. In this regard, the essential parameters in the lag method were speed limit, crossing size, time to turn, lag length, and distance from the downstream signal, which was identified as being substantial. Similarly, in the method for gap acceptance, median form, distance from the downstream signal, time to turn, overall wait time, and gap length were identified as substantial. As a result, critical lag and gap were discovered to be different over an extensive range of parameters based on the junction type. Furthermore,

Devarasetty et al. [23] found that the value of the critical lag is more significant than the critical gap value.

Devarasetty et al. [23] demonstrated that HCM utilizes an individual critical gap value for all types of junctions. Nevertheless, in this investigation, the critical gap value was identified as a variety and was influenced by the traffic characteristics existing at junctions. Therefore, this method should result in much more precise critical lag/gap values and consequently lead to a more precise evaluation of left-turning operational performance at priority junctions. In addition, Devarasetty et al. [23] pointed out that additional investigations need to be carried out to calibrate the values of coefficients for the improved gap/lag acceptance method. This consists of additional field and traffic parameters, including junction geometry, pedestrian activity, and heavy vehicle traffic.

*3.14. McGowen's Method*

Current techniques for calculating critical gaps need a range of data for accepted and declined gaps for each vehicle. An alternative method was created by McGowen and Stanley [56] that could calculate values of the critical gap using information including only declined or accepted gaps. Based on this study, the capability for critical gap computation utilizing declined gap data only is much more beneficial for the following reason: vehicles on a major road might slow down to create specific room for turning vehicles from a minor or major road. Consequently, the approved gap information might be longer than the gap values the motorist saw. In addition, a comparative analysis between this method and Troutbeck [7] was performed and suggested that this method yields precise values of the critical gap.

To do so, a probability density function for gaps on the main road was assumed as a negative exponential distribution. In this case, Khattak and Jovanis [57] found that using this distribution was not a problem, especially when traffic flows were high. In addition, using the composite distribution recommended by May [58] created an excellent fit during greater traffic flow. McGowen and Stanley [56] listed some advantages and disadvantages of the above method as follows:

1.  The Troutbeck technique needs to utilize all accepted and rejected gaps for calculating the critical gap. Nevertheless, the significant advantage of utilizing the suggested method is that it allows for the use of accepted or declined gap times only. Consequently, this procedure can be helpful if only a set of data using one kind of gap is available.
2.  The suggested method gives a fair calculation, but based on the simulation results, the Troutbeck method probably has a very small amount of bias.
3.  Based on the suggested method, the major disadvantage is that the gap period distribution follows a particular functional form. Therefore, this is especially difficult when utilizing the declining gap only. In this regard, statistical tests such as the chi-squared need to be conducted to ensure that the values of the gap comply with the assumed distribution. Accordingly, another functional form can be utilized if the gap values do not follow the assumed distribution.
4.  Because the suggested technique needs to determine an assumed distribution of the gap lengths in the traffic flow rate, it might not be suitable for very high or very low traffic flows. Therefore, additional research is needed to identify the variety of traffic flows for which the suggested technique is suitable for an assumed functional form of the gap length distribution.

## 4. Discussion

In the previous section, a range of methods related to the critical gap research were reviewed to be offered as guides and to equally present detailed information related to the critical gap at priority junctions. In this way, it is believed that the delay brought on by a low priority traffic approach will have a significant impact on how well a priority junction performs, where the critical gap is a crucially important objective indicator of a driver's

effectiveness and is used in studies about road safety, delays, and capacities at priority junctions. Moreover, the critical gap is not regular but differs from motorist to motorist as well as from time to time. The critical gap was also discovered to differ with automobile type, approach gradient, junction geometry, delay, and weather conditions.

In this review, a substantial evaluation and discussion concerning the current issues regarding the critical gap, such as their strengths and weaknesses, managed to emerge. This might assist further studies in this field. Thus, a couple of the aforementioned methodologies have significant drawbacks with respect to critical gap analysis at priority intersections. For instance, Ashworth's technique is valid only if minor street vehicles are assumed to arrive at the "Give Way" line independent of the arrival of the following main street vehicle. In Pollatschek's method, the assumption is not correct because of variations among motorists, such as driving manners, age, and gender. The method presented by Yan did not perform field validation for the simulator data. Guo's method needs additional investigation to calibrate the values of coefficients on the improved gap acceptance method, depending on large sample sizes. Lastly, Devarasetty's method needs to be carried out to calibrate the values of coefficients on the improved gap/lag acceptance method, which includes more field- and traffic-specific parameters.

According to the techniques explained in the prior section, numerous methods have demonstrated substantial benefits. Solberg established a method in which lag and gap are able to be used with each other. Pant's method utilized an ANN for junctions in rural and urban regions with approximately low and significant traffic volumes. Tian recognized various elements that might influence the critical gap. Although McGowen calculates values of the critical gap by using only the declined or accepted gap, the gap period distribution follows a particular functional form.

In practice, the most common methods are those of Raff [6] and Troutbeck [7] in unsaturated situations. In addition, Brilon's [21] research identified that Troutbeck's [7] method provides the best outcomes. Therefore, this method was suggested for calculating the critical gaps in TRB [9–11]. Wu [54,55] described how the method of Troutbeck [7] depends on the theory of maximum likelihood evaluation and is a microscopic method. This technique necessitates two assumptions, namely a log-normal distribution for crucial gaps as well as homogeneous and consistent driver behavior. In this respect, declined gaps must be lower than authorized gaps, and only the maximum declined gap and the approved distance between individual cars may be used pairwise. There is no use for data pairs with refused gaps that are more severe than permitted gaps. In some instances, more than fifty percent of the computed gaps cannot be utilized. This is a tremendous waste of data gathered. In addition, Troutbeck's approach is quite intricate and yields weak results. For this procedure to provide consistent findings, a large sample size is also required.

Considering the aforementioned shortcomings, the size of the critical gap was evaluated utilizing Wu's [54,55] equilibrium of probability technique. The theoretical background of this new method is the probability equilibrium between the declined and approved gaps. The equilibrium is founded macroscopically on the cumulative distributions of the declined and approved gaps. It turns out that the method derived via the macroscopic equilibrium is actually additionally appropriate regarding computing critical gaps. If the same sample sizes are applied, Wu's new approach produces equivalent results to Troutbeck's method. It has a solid theoretical foundation (regarding the Markov chain and probability equilibrium) and resilient results. In addition, it is devoid of any methodological assumptions. It requires neither a fixed distribution function of crucial gaps nor the uniformity or consistency of driver behavior. The empirical cumulative distribution function (CDF) of the critical gap may be calculated using Wu's technique, which takes into account all suitable gaps (as opposed to merely the largest decreased gaps as in the Troutbeck method). In general, Table 1 detail the benefits and limitations of the approaches listed. Figure 2 is a flowchart illustrating the technique for analysis and comparison.

**Table 1.** Summary of critical gap methods at priority junction.

| Source | Method | Advantages (A)/Limitations (L) |
|---|---|---|
| Raff's method | • Cumulative probabilities of approved and declined lag. <br> • Extended through taking into consideration solely gaps or from mixing lags and gaps with other researchers. | • Slight bias whenever the volume of traffic is low (A). <br> • By using solely lag data, the research uncovered a considerable inaccuracy in prediction when using this method (L). |
| Solberg's method | • Utilized a probit evaluation to establish a statistical procedure. | • Lag and gap are able to be used with each other (A). |
| Ashworth's method | • Included both gap and lag size. <br> • Assumed to comply with an exponential distribution. | • Valid only if minor street vehicles are assumed to arrive at the "Give Way" line independent of the arrival of the next main street vehicle (L). |
| Adebisi's method | • The CHOMP computer program was used. | • Only unsaturated situations on a minor street were investigated (L). <br> • Concentrated on left-turning minor street traffic (right-hand driving system), with small or no queue development (L). |
| Troutbeck's method | • Used the maximum likelihood technique. | • This method is suggested for calculating the critical gaps in HCM. <br> • Needs a significant sample size for creating stable outcomes (L). |
| Pant's method | • Binary-logit technique and an artificial neural network (ANN) technique were used. | • ANN would be utilized for junctions in rural and urban regions with approximately low and significant traffic volume (A). |
| Tian's method | • Utilized linear regression technique. | • Recognized various elements that might influence critical gaps (A). |
| Pollatschek's method | • A microscopic decision model. | • Assumption is not correct because of variations of motorists, such as driving manners, age, and gender (L). |
| Yan's method | • Utilized the UCF (University of Central Florida). | • Authors did not perform field validation for the simulator data (L). |
| Sangole's method | • Utilized ANFIS in MATLAB. | • Considered the effect of the behavior of two-wheelers (A). <br> • This study concentrated only on right-turning maneuvers from minor roads (L). |
| Guo's method | • Based on the probability function of the declined and the approved gap, critical gaps are theoretically acquired. | • Additional investigation is required to calibrate values of coefficients on the improved gap acceptance method depending on large sample sizes (L). |
| Wu's method | • Based on probability equilibrium in between gaps of declination and acceptance. | • This method yields results similar to those from Troutbeck's model (A). <br> • This process can be quickly carried out in a spreadsheet (A). <br> • It needs neither the predetermined distribution function of the critical gaps nor homogeneity or consistency (A). |

**Table 1.** *Cont.*

| Source | Method | Advantages (A)/Limitations (L) |
| --- | --- | --- |
| Devarasetty's method | • Binary logit method was used. | • Needs to be carried out to calibrate values of coefficients on the improved gap/lag acceptance method consisting of more field- and traffic-specific parameters (L). |
| McGowen's method | • Probability density function for gaps on the main road was assumed as a negative exponential distribution. | • Calculated values of the critical gap by using information including only declined or accepted gaps (A).<br>• The gap period distribution follows a particular functional form (L). |

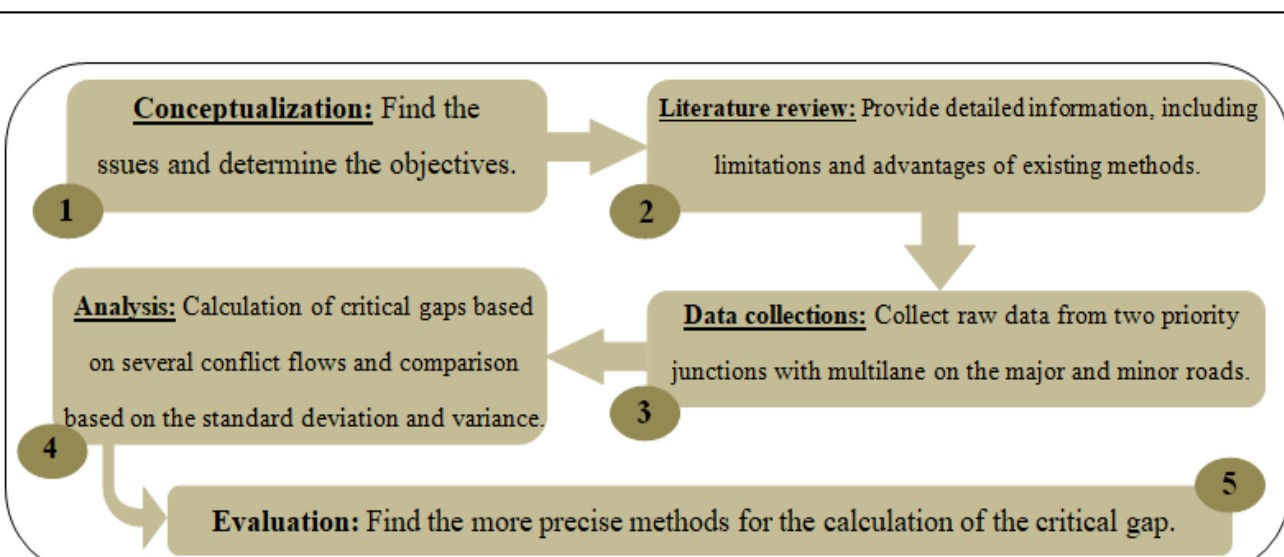

**Figure 2.** Flow chart of the study.

## 5. Critical Gap Assessment

### 5.1. Data Gathering and Site Description

The essential information needed for this research is the entire set of accepted and rejected gaps for vehicle movement from the minor street as well as traffic flow or the volumes of both the minor and major approaches. It is recognized that a relatively precise measurement of motorists' critical lag or gap may be acquired from numerous site studies and the enormous amount of gap approval and rejection information. Nevertheless, due to the limitations of resources and time, the amount of information gathered for this specific research has to be compromised among a realistic, reasonable attempt at data collection and the requirement for sufficient information for statistical evaluation.

Numerous observations were created of the various junctions all over urban and suburban site studies. The purpose was to determine an appropriate field study for data collection reasons. The selection of locations to be examined depends on the subsequent specifications:

1. Outstanding overhead vantage angles for video filming;
2. Great accessibility and security for the enumerators and devices throughout the data collection procedure;
3. Excellent sight distances to ensure that sight distances do not interfere with the interactions between drivers;
4. Appropriate traffic volumes on both main and minor streets to acquire an excellent quality of information.

Unfortunately, site studies that possess all the mentioned above criteria were hard to find. Subsequently, the priority junction selected regarding this particular investigation was a compromise among the criteria offered previously [59]. The current research chose two priority junctions situated in a central business district (CBD) region in Johor Bahru, Malaysia. Figure 3b displays the traffic lane configuration at priority junctions. These site studies were chosen simply because the initial short traffic counts revealed acceptable amounts of turning movements suitable for the goals of the field observations.

In this research, field information collection was accomplished by utilizing a video camera technique. Ashworth [60], Huang [61], Ke et al. [62], and Sahraei et al. [63] explained the positive aspects of applying a video recording procedure for traffic information gathering. The mentioned technique has also been utilized in numerous critical gap and delay studies, e.g., Ashalatha and Chandra [22], Sahraei and Akbari [64], Vinchurkar et al. [65], and Deepthi and Ramesh [66]. Figure 3a shows the locations of cameras at priority junctions.

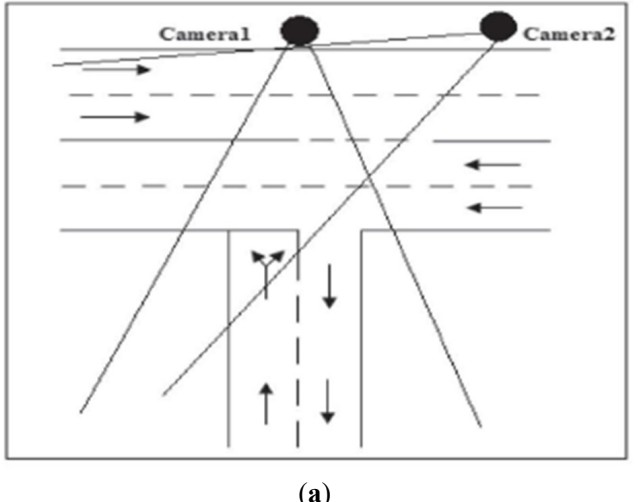

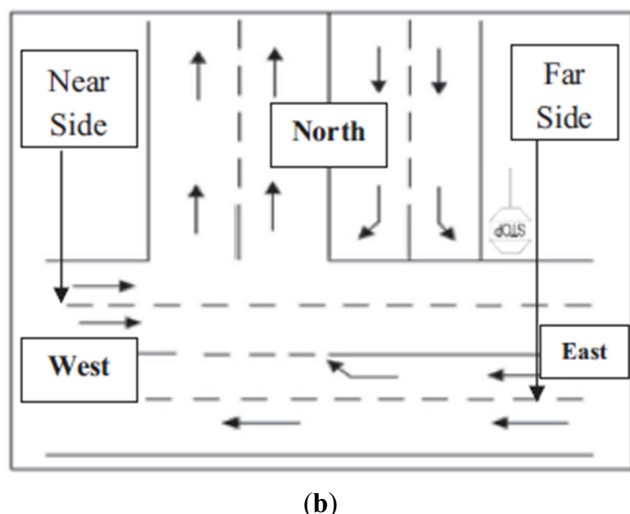

(**a**)  (**b**)

**Figure 3.** (**a**) Position of cameras. (**b**) Traffic lane arrangement at priority junctions.

A total of 120 h of recording time at two junctions was adopted for the field study. The recording periods were conducted for around 12 h (i.e., 7:00 am–7:00 pm) for 5 working days per week. Based on a preliminary analysis, these recording times were evaluated as appropriate regarding examining the required traffic parameters within a variety of traffic flows.

Each of the recordings comprising the recorded scenes was played back many times to collect the data, including traffic volume on the minor and major streets as well as accepted and rejected gap times for vehicle mobility from the minor towards the main street. A personal-computer-based event recorder was used to obtain the information necessary to determine the above information through the footage. By manually counting (i.e., using a counter) the number of cars traveling in each direction at the intersections, it was possible to estimate the traffic flow rates on the minor and main routes. During this procedure, the laptop played the gathered data in fast-forward mode. A stopwatch was used to gather the magnitudes of declined and approved gaps regarding minor roadways for both left and right turns. Similarly, the time period length (based on seconds) was recorded among cars on a main street that a vehicle on a minor street was willing to travel. During this procedure, the footage was seen in real-time. The vast majority of these departure and arrival time data were retrieved using the same time reference for all traffic directions. This was essential because almost all events have to be organized correctly depending on the individual occurring times for gap acceptance evaluation.

### 5.2. Calculation and Comparison of Critical Gap Methods

The vast majority of previously explored methodologies were employed to evaluate the critical gap at the priority junction, which depends on the information of several field investigations. More importantly, it needs to be mentioned that the value of the critical gap that is reliant on various techniques was computed depending on the associated input variables, such as conflict traffic volume on the main street and traffic volume on the minor street for each subject movement, as well as rejected and approved gaps. Whenever additional input factors for a specific method were necessary, these variables were collected from the site research. For example, Troutbeck's method needs the greatest rejected gap as well as the accepted gap of individual cars. The observed data of the variables (i.e., for right and left turns of a minor street, separately) were fed into the existing techniques, as examined previously, in calculating the critical gap for the objective of investigating the methods' goodness of fit.

The overall number of approved and denied gaps regarding a right turn from a minor road at junctions including a four-lane main/four-lane minor street was 9785 and 1745, respectively, according to this study. At the identical intersections, these figures represented 3974 approved and 638 rejected gaps regarding a left turn of a minor street.

The computed critical gaps that rely on formulas were then compared depending on the estimated variance and standard deviation for each method, as shown in Table 2. In statistics and probability theory, variance (Equation (2)) is actually the requirement of the squared deviation of a random parameter from its mean. It calculates how far a set of quantities is dissipated from their average magnitude, and it is the square of the standard deviation.

$$\text{Var}(x) = \frac{1}{n}\sum_{i=1}^{n}(x_i - \mu)^2 \qquad (2)$$

**Table 2.** Values of critical gaps, variance, and standard deviation for several methods.

| Methods | Left Turn | | | | | | Right Turn | | | | | |
|---|---|---|---|---|---|---|---|---|---|---|---|---|
| | Conflict Flow (veh/h) | | | | * Var | * StDv. | Conflict Flow (veh/h) | | | | * Var | * StDv. |
| | 500 | 1000 | 1500 | 2000 | | | 500 | 1000 | 1500 | 2000 | | |
| Raff | 2.8 | 2.83 | 2.92 | 3.05 | 0.013 | 0.115 | 3.9 | 3.97 | 4.01 | 4.17 | 0.014 | 0.118 |
| Solberg | 4.4 | 4.45 | 4.63 | 4.78 | 0.027 | 0.165 | 5.6 | 5.58 | 5.76 | 5.94 | 0.032 | 0.178 |
| Ashworth | 3.7 | 3.88 | 3.98 | 4.07 | 0.024 | 0.154 | 4.8 | 5.04 | 5.14 | 5.19 | 0.025 | 0.159 |
| Adebisi | 2.9 | 2.96 | 3.1 | 3.25 | 0.023 | 0.153 | 4.1 | 4.16 | 4.28 | 4.47 | 0.030 | 0.173 |
| Troutbeck | 2.9 | 2.88 | 2.95 | 3.07 | 0.010 | 0.098 | 3.9 | 4.02 | 4.08 | 4.14 | 0.010 | 0.098 |
| Pant | 2.7 | 2.91 | 2.95 | 3.18 | 0.034 | 0.185 | 3.9 | 4.07 | 4.17 | 4.29 | 0.033 | 0.182 |
| Tian | 3.8 | 3.81 | 3.95 | 4.1 | 0.021 | 0.144 | 4.9 | 5.03 | 5.13 | 5.27 | 0.020 | 0.142 |
| Pollatschek | 3.1 | 3.26 | 3.39 | 3.46 | 0.020 | 0.142 | 4.4 | 4.41 | 4.57 | 4.68 | 0.021 | 0.144 |
| Yan | 3.4 | 3.73 | 3.75 | 3.84 | 0.032 | 0.178 | 4.5 | 4.94 | 4.86 | 4.97 | 0.045 | 0.212 |
| Sangole | 2.9 | 3.06 | 3.15 | 3.26 | 0.022 | 0.148 | 4.1 | 4.28 | 4.29 | 4.43 | 0.021 | 0.144 |
| Guo | 3.5 | 3.62 | 3.75 | 3.92 | 0.028 | 0.166 | 4.7 | 4.83 | 4.87 | 5.06 | 0.023 | 0.153 |
| Wu | 2.8 | 2.82 | 2.94 | 3.03 | 0.011 | 0.105 | 4 | 4.02 | 4.09 | 4.17 | 0.007 | 0.083 |
| Devarasetty | 3.5 | 3.52 | 3.55 | 3.83 | 0.028 | 0.167 | 4.6 | 4.69 | 4.72 | 4.96 | 0.026 | 0.160 |
| McGowen | 2.9 | 2.95 | 3.13 | 3.19 | 0.019 | 0.136 | 4.1 | 4.13 | 4.27 | 4.38 | 0.020 | 0.143 |

* Var: variance; StDv.: standard deviation.

Variance and the standard deviation are excellent ways to discover all feasible magnitudes (i.e., estimated critical gaps) that can be taken within a given range. Accordingly, a considerable variance and standard deviation imply that the quantities in a set are generally far from the mean and each other. A tiny variance implies that the quantities are generally closer together in value.

The comparative evaluations provided in Table 2 indicate that some of the assessed techniques are suitable for calculating critical gaps at priority junctions, although most did not present a suitable value for critical gap evaluation. For instance, the magnitude of variance for Raff's techniques regarding right and left turns from the minor street was actually discovered to be around 0.01, while it was estimated for Ashworth's method to be around 0.025.

In general, it can be concluded that the values of variance for Troutbeck's, Wu's, and Raff's methods were close to the mean regarding right and left turns from the minor road. In addition, among the techniques compared in this research, McGowen, Tian, Pollatschek, and Sangole were also capable of presenting a strong estimate of the approximate magnitude of the critical gap regarding both sides of the minor road at priority junctions. Apart from the above methods, Yan's and Pant's methods could not precisely estimate the magnitude of the critical gap, where the variance for Pant's and Yan's methods to be estimated was more than 0.033 and 0.045, respectively.

## 6. Conclusions

The critical gap is actually one of the necessary variables for calculating the capacity and delay of priority junctions that cannot be immediately calculated on site. This paper's extensive analysis of the limitations and merits of current critical gap approaches may contribute to the advancement of research on this subject. Therefore, a comprehensive assessment and comparison among existing methods for estimating critical gaps at priority junctions were carried out. The precision of the critical gap calculation techniques was examined utilizing factual information with a total of a 120 h recording period from two junctions, particularly multilanes on main and minor streets. The recording periods were conducted from 7:00 am to 7:00 pm for five working days per week. The current research managed to determine that Troutbeck's, Wu's, and Raff's techniques offered consistent results, whereas the critical gap estimated by Yan's and Pant's techniques was discovered to differ with the magnitude of variance and standard deviation. In addition, the results of this research demonstrated that McGowen's, Tian's, Pollatschek's, and Sangole's methods were also able to offer a suitable estimate of the approximate magnitude of the critical gap regarding both sides of the minor street at priority junctions. As the study limitations demonstrate, only two sites with multilanes on the main and minor streets were addressed for validation. In addition, both junctions were considered flat without any impact of an upstream/downstream traffic signal on capacity at the selected TWSC junctions. Thus, for future studies, it is suggested that more site studies with different geometries and traffic volumes (i.e., flat or with different grades and taking into account the effect of a traffic signal upstream or downstream) be used for testing and validating, even though some techniques did not give good results or gave a large amount of variance and standard deviation. Additionally, for future work, more methods, even very old ones, need to be added so that this reference is as complete as possible. In addition, the critical gap calculation must be taken into account for staggered and crossroad junctions.

**Funding:** This research received no external funding.

**Institutional Review Board Statement:** Not applicable.

**Informed Consent Statement:** Not applicable.

**Data Availability Statement:** Data available on request from the authors.

**Conflicts of Interest:** The author declares no conflict of interest.

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
