# Peer review of "Comparative Investigation of the Critical Gap at Priority Junctions: A Review Paper"

_futuretransp, doi:10.3390/futuretransp3020028_

Round 1
Reviewer 1 Report
The paper deals with an interesting subject involving a review of critical gap calculations at priority junctions. The author needs to elaborate more on the methodological aspect of the paper and on future research directions. Otherwise, the contribution of this paper in terms of summarizing the state-of-the-art appears satisfactory. Specific issues which should be tackled before publication include the following:
- The abstract needs more careful and precise wording so that the reader can understand what exactly the paper involves in order to be stand-alone. Please define whether the investigated gap is purely spatial or spatiotemporal, if it only concerns automobiles (or can be between automobiles and VRUs) and if it is a singular, universal threshold or a function-like approach. In the main manuscript, these issues are clarified appropriately.
- It is critical to add the review strategy of the paper. While a full PRISMA is preferable, at the very least please mention which scientific search engines and which search terms were employed to determine the final studies included in this review. This step ensures that reviews are replicable and it is very important.
- The statement of lines 42-43 needs a supporting reference.
- In section 2, a simple visual representation of the critical gap would be beneficial for the paper.
- The cases of staggered junctions or junctions with non-perpendicular angles should be examined in the study. If there is no literature available for this topics, then a mention should be made.
- Perhaps the author could shorten all subsections of Chapter 3 which contain very outdated references (e.g. subsections 3.1 – 3.8) wherever possible.
- The passage between lines 456-480 is quite vague (e.g. ‘utilizing this distribution was not troublesome’ and ‘the bias… is probably tiny’) and thus needs more precise wording. Please rephrase it.
- It is important to include a section regarding future research directions and suggestions based on the experience that the author acquired from reviewing the present studies.
Author Response
Dear reviewer
First of all, we would like to thank you for your valuable time and for your constructive criticisms and valuable comments, which were of great help in revising the manuscript. Here is a point-by-point response to the comments provided in the email.
Comments:
The paper deals with an interesting subject involving a review of critical gap calculations at priority junctions. The author needs to elaborate more on the methodological aspect of the paper and on future research directions. Otherwise, the contribution of this paper in terms of summarizing the state-of-the-art appears satisfactory. Specific issues which should be tackled before publication include the following:
1- The abstract needs more careful and precise wording so that the reader can understand what exactly the paper involves in order to be stand-alone. Please define whether the investigated gap is purely spatial or spatiotemporal, if it only concerns automobiles (or can be between automobiles and VRUs) and if it is a singular, universal threshold or a function-like approach. In the main manuscript, these issues are clarified appropriately.
- Response:
- Thank you very much for this precise comment. We have rewritten the abstract and added the abovementioned factors to the conclusion. All changes can be seen in the green color.
2- It is critical to add the review strategy of the paper. While a full PRISMA is preferable, at the very least please mention which scientific search engines and which search terms were employed to determine the final studies included in this review. This step ensures that reviews are replicable and it is very important.
- Response:
- To be honest, we're not sure what the comment means, especially when it comes to "scientific search engines and which search terms were employed." We are so sorry for this case. In our opinion, the respected reviewer has asked for related research questions. Therefore, we have provided a robust research question between lines 58 and 63. The green color makes it apparent. We hope that the respected reviewer accepts this correction.
3- The statement of lines 42-43 needs a supporting reference.
- Response:
- Thank you for your careful consideration. The supporting references, including Raff [6], Troutbeck [7], and Guo and Lin [8], were provided for the mentioned sentence. The green color makes it apparent.
4- In section 2, a simple visual representation of the critical gap would be beneficial for the paper.
- Response:
- In response to what the reviewer said, figure 1 shows how the priority junction gap looks. It can be seen by the green color.
5- The cases of staggered junctions or junctions with non-perpendicular angles should be examined in the study. If there is no literature available for this topics, then a mention should be made.
- Response:
- Thank you very much for this comment. The scope of this research was limited to the priority junctions (Two-Way Stop Controlled (TWSC)), i.e., three-legged junctions. For this reason, we provided a comprehensive comparative analysis for this type of junction. With respect to the reviewer’s comment, we have added this fact in the abstract as well as a paragraph in the introduction (lines 65–68). In addition, we have added some sentences in the conclusion related to limitations and factors that must be considered in the future.
6- Perhaps the author could shorten all subsections of Chapter 3 which contain very outdated references (e.g. subsections 3.1 – 3.8) wherever possible.
- Response:
- Thank you very much for this strong review. Because the goal of this review paper is to roughly relate all methods over the last 70 years, from Raff's technique in 1950 to the present, some references will be outdated. With respect to the reviewer's comment, we have added some new references in Section 3. The green color makes it apparent. To make the paragraphs shorter, we provided Tables 1 and 2 and summarized all advantages and limitations previously.
7- The passage between lines 456-480 is quite vague (e.g. ‘utilizing this distribution was not troublesome’ and ‘the bias… is probably tiny’) and thus needs more precise wording. Please rephrase it.
- Response:
- Thank you for this comment. The suggested parts were paraphrased. The vague parts were corrected, and the meaning is clear now. The green color makes it apparent.
8- It is important to include a section regarding future research directions and suggestions based on the experience that the author acquired from reviewing the present studies.
- Response:
- Thank you for this comment. Although the future work was embedded in the conclusion, it was again rewritten precisely. Now it can provide suitable directions and suggestions based on the experience analysis. The green color makes it apparent. We hope that the reviewer accepts these changes.
Yours sincerely

Reviewer 2 Report
This paper reviews the development of critical gap assessment technology in the past 70 years. The accuracy of these methods is evaluated utilizing factual information through two priority junctions, i.e. three-legged formed, with multilane on the main and minor street. This paper makes a detailed summary and comparative analysis of the research on critical gap. It provides meaningful guidance for future research.
Author Response
Dear reviewer
First of all, we would like to thank you for your valuable time and for your constructive criticisms and valuable comments, which were of great help in revising the manuscript. Here is a point-by-point response to the comments provided in the email.
Comments:
This paper reviews the development of critical gap assessment technology in the past 70 years. The accuracy of these methods is evaluated utilizing factual information through two priority junctions, i.e. three-legged formed, with multilane on the main and minor street. This paper makes a detailed summary and comparative analysis of the research on critical gap. It provides meaningful guidance for future research.
- Response:
- Thank you very much for your description and the positive side of the manuscript.
Yours sincerely

Reviewer 3 Report
This paper provides a comprehensive review of the current research status of the techniques established over the past decades from Raff's technique till now. The accuracy of these methods is evaluated utilizing factual information through two priority junctions multilane on the main and minor street.
I think the authors of this paper provided a very complete research summarization. But there are some minor grammatical errors can be found in this manuscript. Please consider a very careful proofreading before final revision.
Author Response
Dear reviewer
First of all, we would like to thank you for your valuable time and for your constructive criticisms and valuable comments, which were of great help in revising the manuscript. Here is a point-by-point response to the comments provided in the email.
Comments:
This paper provides a comprehensive review of the current research status of the techniques established over the past decades from Raff's technique till now. The accuracy of these methods is evaluated utilizing factual information through two priority junctions multilane on the main and minor street.
- I think the authors of this paper provided a very complete research summarization. But there are some minor grammatical errors can be found in this manuscript. Please consider a very careful proofreading before final revision.
- Response:
- A native English speaker checked and proofread the manuscript. It is highlighted in green throughout the manuscript.
Yours sincerely

Reviewer 4 Report
This manuscript is well-written, and the topic is valuable. There are some minor comments as follows:
- The introduction must be organized better. Especially, research questions should be stated clearly.
- In Section 3, some of the sub-sections have only one reference. In my opinion, the authors should improve the references of the study critically.
- The future directions require improvement. In my opinion, adding more studies cannot be enough for future directions.
Author Response
Dear reviewer
First of all, we would like to thank you for your valuable time and for your constructive criticisms and valuable comments, which were of great help in revising the manuscript. Here is a point-by-point response to the comments provided in the email.
Comments:
This manuscript is well-written, and the topic is valuable. There are some minor comments as follows:
1- The introduction must be organized better. Especially, research questions should be stated clearly.
- Response:
- Thank you very much for your comments. In order to improve the introduction, three research questions and aims were added. Its green color makes it obvious.
2- In Section 3, some of the sub-sections have only one reference. In my opinion, the authors should improve the references of the study critically.
- Response:
- We tried to find original references for each method because this is a review paper. As you can see from the green color, we have added some new references for each method in response to the reviewer's comment. But we could not find more references for some methods like “3.6. Pant’s Method” and “3.13. Devarasetty’s Method,” because they are not very popular in comparison with other methods.
3- The future directions require improvement. In my opinion, adding more studies cannot be enough for future directions.
- Response:
- Thank you for this comment. The part about future work was rewritten, and more items were added. We hope that the reviewer accepts these changes.
Yours sincerely

Round 2
Reviewer 1 Report
The authors have taken into account all of the previous comments and they have addressed them to a very satisfactory degree by rewriting major sections of the paper. The paper appears ready for publication and could therefore be accepted.